# Graphene-Induced Performance Enhancement of Batteries, Touch Screens, Transparent Memory, and Integrated Circuits: A Critical Review on a Decade of Developments

**DOI:** 10.3390/nano12183146

**Published:** 2022-09-10

**Authors:** Joydip Sengupta, Chaudhery Mustansar Hussain

**Affiliations:** 1Department of Electronic Science, Jogesh Chandra Chaudhuri College, Kolkata 700033, India; 2Department of Chemistry and Environmental Science, New Jersey Institute of Technology, Newark, NJ 07102, USA

**Keywords:** graphene, nanomaterial, transparent electronics, flexible electronics, touch screen, field effect transistor

## Abstract

**Highlights:**

**Abstract:**

Graphene achieved a peerless level among nanomaterials in terms of its application in electronic devices, owing to its fascinating and novel properties. Its large surface area and high electrical conductivity combine to create high-power batteries. In addition, because of its high optical transmittance, low sheet resistance, and the possibility of transferring it onto plastic substrates, graphene is also employed as a replacement for indium tin oxide (ITO) in making electrodes for touch screens. Moreover, it was observed that graphene enhances the performance of transparent flexible electronic modules due to its higher mobility, minimal light absorbance, and superior mechanical properties. Graphene is even considered a potential substitute for the post-Si electronics era, where a high-performance graphene-based field-effect transistor (GFET) can be fabricated to detect the lethal SARS-CoV-2. Hence, graphene incorporation in electronic devices can facilitate immense device structure/performance advancements. In the light of the aforementioned facts, this review critically debates graphene as a prime candidate for the fabrication and performance enhancement of electronic devices, and its future applicability in various potential applications.

## 1. Introduction

From a historical perspective, graphene was theoretically predicted as a part of graphite in the 1940s [1] and, in 1962, a single layer of graphite was first experimentally observed by Boehm et al. [2]. In 1994, Boehm coined the term “Graphene” [3]. Nearly two decades later, Novoselov et al. [4] reported, in the journal Science, the observation of graphene. Subsequently, graphene became the pivotal subject of research in the scientific community, which resulted in the rapid development of this field. The ultra-thin (0.35 nm) and super-light honeycomb configuration of graphene, with a planar density [5] of 0.77 mg/m^2^, captured great interest, owing to its novel nanostructure with unique physical and chemical properties [6]. In terms of crystal structure, graphene is the strongest material on earth [7], as well as being flexible [8]. This conclusion is based on the value of its Young’s modulus (1.0 terapascals), its intrinsic strength (130 gigapascals), and its breaking strength (42 N/m) [9]. Graphene exhibits supreme thermal conductivity [10], having a value up of to 8000 W/m·K [11], with a huge specific surface area of 2630 m^2^/g [12]. The transparency of graphene is another peerless feature, in that the transparency of a single layer of graphene is nearly 98% under visible light [13]. Its ultrahigh room-temperature electron mobility of 2 × 10^5^ cm^2^V^−1^S^−1^ [14] established graphene as the most conductive material at room temperature to date, with a conductivity of 1.42 × 10^6^ S/m [15] and a sheet resistance of 125 Ω/sq [16]. Among these fascinating properties, some handpicked properties can be stitched together for specific applications in electronic devices, as shown in Figure 1. The incorporation of graphene in electronic devices enhances their performance enormously.

## 2. Graphene Structure and Properties

Graphene is an atomically thin two-dimensional allotrope of carbon, comprising a single sheet of sp^2^-hybridized carbon atoms, where one atom forms each vertex. Here, the σ-bond is formed by the combination of S, Px, and the Py orbital, while the Pz orbital creates the π-bond. The π bond is mostly responsible for the large electrical conductivity of graphene [17]. Scanning tunnelling microscopy (STM) topography of graphene with a bond length of 1.4 Å [18] is shown in Figure 1 (top left), revealing an atomically resolved graphene lattice where the crystal symmetry is created by two equivalent carbon sublattices, A and B. Similar results were obtained from a transmission electron microscopy (TEM) study, shown in Figure 1 (top right).

There are three kinds of arrangements of carbon chain at the edge of graphene, namely zigzag, armchair, and arbitrary fashion, shown in Figure 1 (bottom left). The conduction behavior of graphene depends on the type of carbon chain arrangements at the edges. Zigzag arrangements of the carbon chain are responsible for metallic conduction, but a nanoribbon with armchair arrangements of the carbon chain acts either as a metal or a semiconductor [23]. Moreover, the bandgap of graphene nanoribbons varies inversely with the width of the nanoribbon [24]; this facilitates bandgap engineering for the potential application of graphene in nanoelectronic devices.

Moreover, graphene is so transparent that clear discrimination of single, bilayer, and multiple-layer graphene is possible by using contrast spectra; these can be generated from the reflection of a white light source from graphene, placed on an Si substrate with a 285 nm SiO_2_ capping layer [25]. An illustration of the variation in graphene transparency with the change in thickness is shown in Figure 1 (bottom right). Thus, theoretical and experimental results reveal that by varying the graphene thickness, the optical property of graphene can be controlled.

In the case of graphene, the first Brillouin zone contains four unique points in reciprocal space, including the Γ point at the center of the cell, K and K′ at inverse corners (Dirac points), and M at the edge center, the midpoint between K and K′ (Figure 2a). Monolayer graphene exhibits Dirac-type linear dispersion near E_F_ = 0 due to its special hexagonal lattice structure, as shown in Figure 2b. The surface and contour plots of energy dispersion are also shown in Figure 2c,d, respectively. The main feature of the energy dispersion of graphene is the six K points at the corners of the Brillouin zone, where the conduction and valence bands meet so that the bandgap is zero only at these points. Interestingly, the shape of the graphene valence and conduction bands resembles cone-like structures, popularly known as the “Dirac cone”, as they meet at the Dirac point (Figure 2e). The band position can be shifted by electron or hole doping [26], thus the Fermi level (E_F_) is displaced to higher or lower energy, respectively, with reference to the Dirac point.

Moreover, similar to a single layer, although a pristine bi-layer does not have a band gap, a bandgap can be created through proper doping [30] (Figure 3a) or by application of an electric field [31] (Figure 3b).

Though graphene was first observed in the 1960s, exact nomenclature and proper classification remain a barrier to a precise definition of different forms of graphene, which acts as a hurdle for its standardization. To overcome this problem, several initiatives were undertaken across the globe. In 2014, Wick et al. [34] proposed a classification grid (Figure 4) for the labeling of different types of graphene, based on three basic properties (i) number of graphene layers, (ii) average lateral dimension, and (iii) atomic carbon/oxygen ratio.

Later, in 2017, a detailed guideline for the nomenclature and standardization of graphene was proposed by the International Organization for Standardization (ISO) [35]. The main four segments of classification were based on (i) terms related to materials (nanoplate, nanoribbon, single/bi/tri-layer graphene etc.), (ii) terms related to methods for producing 2D materials (chemical vapor deposition, roll-to-roll production, mechanical exfoliation, chemical synthesis etc.), (iii) terms related to methods for characterizing 2D materials (scanning-probe microscopy, scanning electron microscopy, transmission electron microscopy, Raman spectroscopy etc.), and (iv) terms related to 2D materials characteristics (defect, grain boundary, stacking, doping etc.). The ISO further expanded the categorization for standardization in 2019 [36]. Recently, the Novoselov group [37] reviewed the global status of the standardization of graphene and concluded that interaction between the scientific community, business community, government, and the general public is necessary for the development of proper categorization, classification, nomenclature and, ultimately, standardization.

The fascinating features of graphene make it ideally suited for further miniaturizing electronics to form ultra-small devices and components for future semiconductor technology. However, for the practical realization of the potential of graphene, it must be synthesized in such a manner that the resulting product is almost defect-free. To date, several methods were devised to fabricate good quality graphene; these can be broadly categorized into top-down and bottom-up approaches. As the names suggest, in the case of the top-down approach, the initial material is a piece of graphite (in general) which is later reduced to single or few-layer graphene by various means (e.g., exfoliation etc.), whereas, in the case of the bottom-up approach, carbon molecules are assembled one by one (e.g., CVD etc.) to prepare a graphene layer. Since this review aims to encompass, primarily, the application of graphene in electronic devices, a detailed description of the synthesis processes is beyond our scope; however, there are many resources [38] available for an in-depth description of the said processes. Moreover, a schematical description of different synthesis methods, quality of grown samples, and their target application fields are presented in Figure 2.

Graphene grown by a suitable synthesis procedure has enormous potential for use in diverse fields, such as flexible electronics [39], conductive inks to print electronic circuits [40], energy storage [41], anti-corrosion [42], biomedicine [43], aerospace [44], etc. However, within the scope of this review, the application of graphene in energy storage, touch screens, flexible electronics, and transistor fabrication are discussed hereunder.

## 3. Application of Graphene in Electronic Devices

### 3.1. Graphene in Batteries

The battery, a device that converts chemical energy to electrical energy, is the utmost requirement for the smooth running of most daily-life hi-tech applications, for example, an electric razor. In the modern era, there is a need for a secondary-type battery that can be charged, discharged into a load, and recharged many times, as opposed to a primary-type battery, which is initially fully charged and must be discarded after use. A lightweight, durable secondary battery with a high energy-storage capacity, and a small charging time, is a prerequisite of modern electronic and electrical gadgets. Since the inception of research on secondary-type batteries [45], the lithium-ion battery (LIB) attracted major attention due to its unmatchable combination of high energy and power density (both gravimetric and volumetric) with long service life, environmental friendliness, and commercial viability [46]. However, continuous performance improvement is required for the sustainable development of any technology. A report [47] indicated that conduction is a key obstacle to the further upgrading of Li-ion batteries. In principle, the transport of Li ions between the anode and cathode materials, via the insertion/deintercalation process, is responsible for the charging and discharging of an LIB (Figure 5). The speed of transportation of Li ions between the electrodes through the electrolyte essentially determines the power capability of an LIB.

Thus, a nanostructure of a large surface area with high electrical conductivity is the key element for creating a high-power LIB. Graphene is the best-fit material to meet these requirements. However, the application of pristine graphene as the anode material presented a stability problem [49]. To circumvent this problem, hybrid materials were created and discovered to be useful for achieving battery enhancement. In the case of basic lithium-ion batteries, graphene can be incorporated either in the cathode or anode. Graphene and its derivatives, along with graphene-based composites, are used as anode materials in basic lithium-ion batteries.

A comparison of electrochemical properties of graphene nanosheet (GNS), expanded graphite, and graphite, revealed that the reversible capacities of GNS are almost double those of expanded graphite electrodes, and three times those of graphite electrodes [50]. Simon et al. [51] used GNS derived from untreated natural graphite and used in LIB, which revealed that the specific capacity in GNS remains higher during the entire cycling process (770 mA·g^−1^ at 100 cycles), which was twice that of the graphite. Yoo et al. [52] prepared GNS materials by chemical exfoliation and used a reassembling process to achieve layered nanosheet products containing several graphene sheets, with varied inter-layer spacing and morphology. They also incorporated carbon nanotubes (CNTs) and fullerenes (C_60_s) in the reassembled structure and found the specific capacity of GNS, GNS + CNT, and GNS + C_60_ to be 540 mAh/g, 730 mAh/g, and 784 mAh/g, respectively. The authors concluded that the difference in electronic structure between graphite and GNS, together with the creation of additional accommodation sites of lithium ions, was responsible for the improved performance. Three types of GNS of varying sizes, edge sites, defects, and layer numbers were fabricated by Li et al. [53]; they demonstrated that a smaller size, increased edge sites, higher defects, and a lower GNS layer number provided a better lithium storage capacity. They found that hydrothermal treatment of GNSs could produce a high reversible discharge capacity of 1348 mAh/g. Lian et al. [54] pointed out that, the large hysteresis and the initial irreversible capacity were the main drawbacks of using GNS in commercial batteries. Adell et al. [55] studied the effect of the specific surface area of graphene nanoplatelets (GNPs) on the capacity of an LIB by preparing GNPs with three different specific surface areas (GNP1: 296 m^2^/g, GNP2: 470 m^2^/g, and GNP3: 714 m^2^/g) Their results revealed that a larger specific surface area was beneficial in enhancing electrochemical performance.

Wu et al. [56] addressed the issue of achieving high power concurrently with a large energy capacity, at faster charge and discharge rates. They prepared heteroatom (N, B)-doped chemically derived graphene and found that doped graphene electrodes exhibited a very high capacity: 1043 mAh/g for N-doped graphene and 1549 mAh/g for B-doped graphene (Figure 6). Moreover, the charge–discharge process of doped graphene is very quick, thus retaining a high-rate capability and excellent long-term cyclability, simultaneously. Zhan et al. [57] prepared free-standing fully fluorinated graphene nanosheets with a high capacity, high-rate capability, and stable cycling performance. Fang et al. [58] synthesized novel mesoporous graphene nanosheets and used them in LIB for an excellent reversible capacity of 1040 mAh/g at 100 mA/g. However, although the rapid charge–discharge process makes the thermal management of LIBs crucial [59], this can be optimized by using a hybrid-phase change material with graphene fillers [60], where the high thermal conductivity of graphene can be exploited.

Along with graphene and its derivatives, graphene-based composites are also used as anodes in LIBs. An MoO_2_/graphene composite was prepared by Bhaskar et al. [61] using a one-pot solution method for application in Li-ion batteries as a high-performance anode material. The homogeneous composite revealed a high conductivity, facilitating fast charge transfer and transport between the oxide nanoparticles and graphene. The battery showed high-rate capability and excellent cycling stability (550 mAh g^−1^ reversible capacity retained after 1000 cycles). Cai et al. [62] prepared iron oxide fibre/reduced graphene oxide (rGO) composites with a capacity of 1085.2 mAh/g at 0.1 A/g, and a cycle life (407.8 mAh/g at 5 A/g for 1500 cycles) with good coulombic efficiency (99%). An innovative approach was developed by Zeng et al. [63] to prepare a nitrogen-doped porous Co_3_O_4_/anthracite-derived graphene (Co_3_O_4_/AG) nanocomposite, via a combination of self-assembly and a heat-treatment process. The uniform distribution of Co_3_O_4_ nanoparticles on the porous graphene helped to achieve a high specific surface area (120 m^2^·g^−1^), and accommodated the high level of nitrogen doping (5.4 at.%). The Co_3_O_4_/AG electrode exhibited a high reversible capacity (845 mAh·g^−1^), outstanding rate capacity (587 mAh·g^−1^), and a reasonable cyclic stability (510 mAh·g^−1^ after 100 cycles). Wu et al. [64] successfully fabricated a composite of hollow Co_3_O_4_ nanocrystals in situ, anchored on holey graphene. This unique design was able to curtail the Li-ion diffusion distance, which, in effect, facilitated Li-ion transport, resulting in a preeminent rate performance (751 mA h/g at 5 A/g), with a high discharge capacity of 1015 mAh/g. Garakani et al. [65] prepared CoCO_3_/graphene aerogel and Co_3_O_4_/graphene aerogel composites, and found that the capacity of both the electrodes exceeded 1000 mAh/g at a current density of 100 mA/g. Moreover, the CoCO_3_/graphene aerogel electrode was better than the Co_3_O_4_/graphene aerogel electrode at low current densities, while the latter electrode showed superior high-rate performance compared with the former electrode.

Choi et al. [66] synthesized a powder consisting of core–shell-structured Ni/NiO nanocluster-decorated graphene, which exhibited better electrochemical properties than those of the hollow-structured NiO-Ni composite and pure NiO powders. The Ni/NiO-graphene composite powder exhibited initial discharge and charge capacities of 1156 and 845 mAh/g, respectively, and the corresponding initial coulombic efficiency was 73% with a good rate performance. Wang et al. [67] synthesized copper ferrites anchored on reduced graphene oxide (CuFeO_2_@rGO and Cu/CuFe_2_O_4_@rGO) and found that the CuFeO_2_@rGO electrode exhibited a high reversible capacity (587 mAh/g after 100 cycles at 200 mA/g) and the Cu/CuFe_2_O_4_@rGO hybrid revealed a superior rate capability (723 mAh/g at 800 mA/g). In both cases, reduced graphene was responsible for the outstanding electron/ion transfer rate and the high specific surface area accommodated the volume expansion of copper ferrites. Lee et al. [68] synthesized MnO_2_/rGO nanocomposites, which showed 222 and 115 mAh/g at a current density of as high as 5 and 10 A/g, and the specific capacitance was well maintained after 400 cycles. Han et al. [69] devised a method to use sulfur as a sacrificial agent in a tin oxide/graphene composite anode and found that the resultant volumetric capacity of graphene–caged tin oxide was 2123 mAh/cm^3^, together with good cycling stability. Wang et al. [70] synthesized a sandwich-like V_2_O_5_/graphene mesoporous composite that exhibited a high reversible capacity of 1006 mAh/g at a current density of 0.5 A/g after 300 cycles. Zhong et al. [71] synthesized Ge–reduced graphene oxide nanocomposites using organic germanium as a precursor: it exhibited a reversible specific capacity of 814 mAh/g, even after 50 cycles at a current density of 0.1 A/g. Li et al. [72] prepared a carbon/graphene double-layer coated-silicon composite (Si/carbon/graphene, Si/C/G) which maintained a specific capacity at 2469 mAh/g and outstanding electrochemical performance in both cycling stability and rate capability. Zhang et al. [73] used a high-energy ball-milling process to produce an SiO_x_/graphene composite with different percentages of graphene. The optimized SiO_x_/graphene composite electrode displayed admirable electrochemical performance, with a reversible specific capacity of 1325.7 mAhg^−1^ and a capacity retention as high as 95.8%. Tang et al. [74] employed the hydrothermal method for the facile synthesis of a silicon-graphene aerogel composite, with a highly dispersible morphology and a 3D porous structure. The specific charge capacity of the composite remained above 1330 mAh g^−1^ at 0.2 A g^−1^, even after 100 cycles. The presence of porosity was identified as the factor responsible for the excellent cycle performance and moderate rate performance during the charge/discharge process. One-step hydrothermal synthesis was employed by Shen et al. [75] to prepare SnO_2_/NiFe_2_O_4_/graphene nanocomposites as anode materials for lithium-ion batteries. In this composite, the SnO_2_/NiFe_2_O_4_ mainly aided in achieving the high capacity, the SnO_2_ smoothened the movement of Li ions, and the graphene provided a high surface that hindered the volume change. The system exhibited a high reversible capacity of 731.5 mA h/g, at a current density of 200 mA/g after 50 cycles.

Graphene-wrapped MnCO_3_/Mn_3_O_4_ nanoparticles prepared by Chen et al. [76], via the one-step method, exhibited extraordinary electrochemical performance, including an outstanding reversible specific capacity (2457.4 mAh g^−1^), an excellent cycling stability (93.3% of the capacity of 2nd cycle after 200 cycles), and a high rate capability (605 mAh g^−1^, even at the high current density of 5 A g^−1^). The lower charge transfer resistance of the composite hindered the occurrence and transmission of cracks during the lithiation-delithiation process and also suppressed the volume change, thereby creating more electrochemically active sites for the Li^+^ ions. Raji et al. [77] addressed a very important LIB issue, known as the dendrite problem [78] (Figure 7). The authors used a seamless graphene–carbon nanotube (GCNT) electrode to reversibly store Li metal with complete dendrite formation suppression. The capacity of GCNT-Li was 3351 mAh/g which is very close to that of bare Li metal (3861 mAh/g).

In the case of metal oxide/graphene composites, the high specific surface area of graphene accommodates the volume expansion of the metal oxide [79], while graphene offers continuous conductive channels for electron/ion transfer with low charge transfer resistance, resulting in improved electrochemical performance of the material. However, because of weak adhesion between graphene layers and active metal oxide particles, the particles tend to aggregate during the process cycle, leading to a large decrease in capacity compared with their first reversible capacity [80]. To overcome this problem Yang et al. [81] devised a new strategy to synthesize graphene-encapsulated metal oxide (GE–MO) by co-assembly between negatively charged graphene oxide and positively charged oxide nanoparticles. The procedure resulted in a GE–MO core–shell structure, where the core comprised the oxide nanoparticles and the shell was flexible and ultrathin graphene. This core–shell structure minimized the aggregation of oxide nanoparticles and exhibited a very high reversible capacity, with excellent cycle performance. This concept was carried forward by Zhou et al. [82] to fabricate other rGO-encapsulated metal oxide composites. They developed a novel scheme based on the ring-opening reaction between epoxy groups and amine groups, and finally electrostatic interaction, to accomplish the self-assembled wrapping of graphene sheets around various metal oxide nanowires and nanoparticles.

Graphene and its derivatives, along with graphene-based composites, are also used as a cathode material in basic LIBs. Some of the most common cathode materials for LIBs are LiCoO_2_ [83], LiMn_2_O_4_ [84], LiFePO_4_ [85], Li_3_V_2_(PO_4_)_3_ [86], and V_2_O_5_ [87], owing to their high capacity, excellent cycle life, thermal stability, environmental benignity, and low cost. However, the low ionic and electrical conductivities of such cathode materials restrict their practical application. Here, graphene plays an important role as an electron-conducting additive for the above-mentioned substances and makes the resulting composite appreciably suitable as a cathode material. Deng et al. [88] used graphene as a conducting additive for LiCoO_2_ cathode materials and found that the surface wrinkles of graphene-wrapped LiCoO_2_ particles formed a plane-contacted interface which enhanced the conductivity of the cathode; the high-rate cycling stability was also improved. The experimental results showed that after 300 cycles, the capacity of a graphene/LiCoO_2_ battery decreased from 145 to 137.8 mAh/g, which was 95.1% of the initial discharging capacity. Pyun et al. [89] found that the discharge capacity, cyclic performance stability, and rate capability of the graphene/LiMn_2_O_4_ electrodes were significantly higher than those of the pristine LiMn_2_O_4_ electrodes. The authors concluded that the high surface area of LiMn_2_O_4_ nanoparticles and good electronic conductivity of graphene were responsible for the enhanced performance. Hu et al. [90] modified the surface of LiFePO_4_ with 2 wt% of electrochemically exfoliated graphene layers, with the resulting composite being able to reach 208 mAh/g in specific capacity, which was beyond the theoretical capacity of 170 mAh/g. They also indicated that such a simple and scalable approach might also be applied to other cathode systems, boosting the capacity of various Li batteries. Rai et al. [91] prepared pure Li_3_V_2_(PO_4_)_3_ nanoparticles and an Li_3_V_2_(PO_4_)_3_/graphene nanocomposite and compared their performance as cathode materials in a basic Li-ion battery. They found that the Li_3_V_2_(PO_4_)_3_/graphene nanocomposite electrode delivered a high reversible lithium storage capacity (189.8 mAh/g at 0.1 C), a superior cycling stability (111.8 mAh/g at 0.1 C, 112.6 mAh/g at 5 C, and 103.4 mAh/g at 10 C after 80 cycles), and a better C-rate capability (90.8 mAh/g at 10 C) in comparison with the pure Li_3_V_2_(PO_4_)_3_ nanoparticle electrode at all investigated current rates. Liu et al. [92] developed a method to integrate graphene into V_2_O_5_ nanoribbons using the sol–gel process; it exhibited a specific capacity of 438 mAh/g, approaching the theoretical value (443 mAh/g), a long cyclability and a significantly enhanced rate capability. Their results indicated that the improved performance originated from the combined effects of graphene on structural stability, electronic conduction, vanadium redox reaction, and lithium-ion diffusion. A unique sandwiched mesostructure was fabricated by Liu et al. [93], where graphene was incorporated between the layers of V_2_O_5_ to form a V_2_O_5_ @graphene@V_2_O_5_ cathode; this demonstrated a full electrode basis capacity ≈ 203 mAh g^−1^ after 2000 cycles. At the same time, they developed a graphene@Si@graphene electrode for use as the anode. They reported that after 200 charge–discharge cycles at 0.4 C, the graphene@Si@graphene mesostructured anode exhibited a capacity of ≈2500 mAh g^−1^. Son et al. [94] fabricated a silica–graphene core–shell structure, named a “graphene ball”, which could be uniformly coated onto a nickel-rich layered cathode; more importantly, the graphene ball itself also served as an anode material. The application of a graphene ball in a full-cell exhibited the possibility of achieving 800 Wh/L in a commercial cell setting, along with a high cyclability of 78.6%, and a capacity retention after 500 cycles at 5 °C and 60 °C. Yong-Jian et al. [95] devised a facile method to prepare 1% concentration graphene slurry with excellent stability, using mechanical agitation, ultrasonic dispersion, and the addition of different dispersants. They employed graphene slurry as a conductive agent in an LIB to improve its electrochemical performance. The first charge/discharge capacity of the system was measured as 1273.8/1723.7 mAh/g, and the Coulomb efficiency was 73.9%, at a 100 mA/g current. Kim et al. [96] devised an advanced energy storage system, the “all-graphene-battery”, whose operation was based on fast surface reactions in both electrodes and resembled both supercapacitors and batteries. The battery delivered a remarkably high-power density of 6450 W/kg, while also retaining a high energy density of 225 Wh/kg. Thus, the “all-graphene-battery” demonstrated that an energy storage system made with exclusively carbonaceous materials was a promising technology for the future. A tabular representation depicting the different electrode materials used for capacity enhancement of LIBs is shown in Table 1.

### 3.2. Graphene Electrodes for Touch Screens

From the inception of the idea of the “touch screen” [97] to date, metal oxides [98] are primarily employed for its production. Among various metal oxides, ITO [99] is the most attractive, owing to its unique combination of visible light transparency and modest conductivity [100]. However, ITO is chemically unstable [101], expensive, and not flexible, and these drawbacks necessitate the search for a better substitute. Subsequently, diverse forms of materials, including nanomaterials [102] underwent feasibility tests in terms of cost, chemical stability, flexibility, transparency, and conductivity. Numerous studies [103,104] demonstrated that graphene had the potential to replace ITO because of its small sheet resistance, large optical transmittance, and the possibility of transferring it onto plastic substrates. Additionally, graphene-based transparent conductive electrodes can be produced on a large scale via the roll-to-roll technique to fabricate touch screens [105]. Even by combining the rod-coating technique and the room-temperature reduction in GO, it is possible to fabricate large-scale highly flexible rGO films directly on PET (polyethylene terephthalate) substrates [106], with low resistance and good transparency (~80%). Ponnamma et al. [107] developed rGO/polydimethylsiloxane (PDMS)-based flexible and transparent capacitive touch-responsive film, where rGO was responsible for sensing. The issue of limited visibility due to the poor dispersion of rGO into the matrix was resolved by the introduction of an ionic liquid and ~70% transmittance was achieved. However, an efficient transfer mechanism [108] and the proper doping of samples with the lateral size of tens of centimeters are the major hindrances to applications of graphene as a touch screen. Guo et al. [109] addressed the proper transfer issue in their research by growing graphene film on copper foil using low-pressure chemical vapor deposition; they transferred the graphene employing a hot-roll-pressing transfer technique, which was later followed by a wet-chemical doping process. The measured value of the sheet resistance of the single-layer graphene reached 200 Ω/sq with an optical transparency of 87.3%; the sheet resistance increased by only ~0.02% after bending, over 10,000 cycles at a radius of 2 mm, which depicted exceptional mechanical flexibility. Large graphene films grown by chemical vapor deposition (CVD) often experience high sheet resistance owing to small grain sizes and high-resistance grain boundaries. Chen et al. [110] devised a method to overcome this issue by employing grain-boundary engineering. They developed a hybrid structure consisting of a network of silver nanowires and a CVD-grown single-layer graphene; the resultant structure exhibited a sheet resistance of 22 Ω/sq at an optical transmittance of 88%. Co-percolating conduction was attributed as the main factor behind the low sheet resistance at moderate nanowire density. In co-percolating conduction high-resistance, graphene grain boundaries are bridged by the silver nanowires, while the high-resistance junctions between the nanowires are bridged by the graphene. The Langmuir-based technique was employed by Jurewicz et al. [111] to fabricate inexpensive transparent electrodes with high conductivity. They deposited graphene on ultra-low-density networks of silver nanowire, thereby creating a network in which inter-wire junctions and adjacent wires were connected via graphene; this resulted in improved electrical properties by several orders of magnitude. They also estimated that the addition of graphene reduced the production cost by more than fifty times with respect to solely silver nanowire-made electrodes. In a similar approach, Zhu et al. [112] used graphene and a metallic grid to form inexpensive, flexible, and transparent conducting films (Figure 8), having a sheet resistance of 3 Ω/square and ~80% transmittance at visible light. The flexibility test revealed that, initially, the conductivity of the hybrid film decreased by 20–30% up to 50 bends; however, the conductivity stabilized thereafter up to 500 bending cycles.

The unique concept of a triboelectric nanogenerator [113] was incorporated into the manufacturing of transparent resistive touch screens by Khan et al. [114] by coupling the triboelectrification with the GFET current. Due to the triboelectric effect, any movement on the screen gives rise to the charges which, in turn, switch on the GFET and modulate current flow thereafter. The low-power devices (180 μW) made with this technology revealed an exceptionally stable touch-sensing performance, with a limit of detection as low as <1 kPa and a response time of ≈30 ms, with stable operation over thousands of cycles. Lee et al. [115] designed a touch-screen sensor using a similar concept. They used graphene as the electrode, PET as substrate, and PDMS as the electrification layer, to develop a touch sensor that provided high electrical conductivity, good optical transparency (~87.8%), and reasonable mechanical stability. The idea of preparing a touch screen using tribotronics was further extended by Tang et al. [116], who formed a touch-free screen sensor to recognize noncontact gestures. They used graphene to fabricate the sensor with 85% transparency. For sustainable development of touch screens, the constituents must be eco-friendly, and graphene-based inks are gaining much popularity in this respect. An environment-friendly, transparent sensor was made by Sadasivuni et al. [117] employing GO-filled cellulose nanocrystals, via the spray method, on polymer substrates. The developed sensor could detect finger movement within a distance of 6 mm, with fast response and recovery times, as well as good stability and high reproducibility. Franco et al. [118] prepared a capacitive multitouch sensing surface employing conductive graphene-based ink and for which carboxymethyl cellulose was used as a binding element. The environmentally friendly graphene-based conductive ink allowed printed line patterns with an electrical resistance of 2.4 kΩ for lines of 0.5 mm in width and five printing steps. A similar material was developed by Tkachev et al. [119] through graphene dispersions using liquid-phase exfoliation and employing an amalgamation of shear-mixing and tip sonication techniques. The developed ink showed an optical transmittance of 78% with a sheet resistance of 290 Ω/square; in addition, there was no significant change in sheet resistance, even after bending with a curvature radius of 28 mm (Figure 9).

### 3.3. Transparent Memory with Graphene

The parametric measures of good electronic gadgets are changing rapidly. The inclusion of multiple functionalities, such as transparency, wearability, flexibility, portability, etc., are the prime aspects of advancement for today’s electronic gadgets, which is leading to a new era of electronic gadgets called “transparent electronics”. Semiconducting nanowires [120] made of wide bandgap semiconductors were initially employed for the fabrication of transparent electronics. It was later found that incorporation of graphene augments the performance of transparent–flexible electronic gadgets because of its finer mechanical properties, negligible light absorbance, and superior mobility. Moreover, because of the high density of states and large work functions with excellent electrical properties, graphene became a promising material for non-volatile transparent, flexible memory devices, in which it can be employed as a conducting channel or a charge-storing layer.

Ji et al. [121] fabricated an array-type organic resistive memory device using graphene electrodes on a PET substrate. The flexible, write-once-read-many (WORM)-type memory device showed a sheet resistance of ~270 Ω/square, with a transparency of 92%. The ON/OFF ratio was ~10^6^ and a retention time of over 10^4^ s was observed during bending cycling up to 10 000 times. A graphene-integrated, highly transparent (transmittance >82%) resistive random-access memory (TRRAM) device was fabricated by Dugu et al. [122]. The device consisted of a hybrid structure of ITO/Al_2_O_3_/graphene and showed stable, non-symmetrical bipolar switching characteristics at low set-reset voltages (<±1 V), with a high on–off ratio of ~5 × 10^3^. The resistive switching mechanism of the device with exceptional endurance and retention characteristics resulted from the formation and rupture of conductive filaments formed due to oxygen vacancies. A flexible and transparent graphene charge-trap memory (GCTM) was fabricated by Kim et al. [123,124] employing a single-layer graphene channel on a polyethylene naphtalate (PEN) substrate. In comparison to the bare PEN substrate, the GCTM lost only 8% transparency and depicted a memory window of 8.6 V, with good retention quality. The stress-based study showed that electrical characteristics were not altered significantly under stressed conditions (Figure 10).

A transparent (90%) non-volatile memory device with large retention times and small programming currents was fabricated by Yao et al. [125] using SiO_x_ and graphene with ON/OFF ratios between 10^3^ and 10^6^. This was a unique amalgamation of graphene properties and well-established SiO_x_ technology for the seamless fabrication of transparent electronics. Yang et al. [126] developed a transparent resistive random-access memory using ZnO and graphene, where graphene acted as a stable resistive element. The fabricated device exhibited good switching characteristics for memory applications. The statistical analysis revealed that the switching yield increased significantly because of the suppression of the surface effect by graphene. Multi-level resistive switching was observed in transparent (above 80%) electronic memory cells fabricated using ITO/rGO/ITO structure by Kim et al. [127]. Good endurance of over 10^5^ cycles was observed in the fabricated device, with long data retention of over 10^5^ s at 85 °C. A complex structure composed of ITO/GO/ITO/PES was fabricated by Wu et al. [128] to achieve flexible and transparent resistive switching memory devices. The resistive switching characteristics depicted that the set and reset voltages were around 1 V and 0.1 V, respectively, and the switching characteristics remained unchanged even after the bending test; the retention time was 10^5^ s. Recently, there is a growing interest among researchers in using graphene–polymer composites in transparent and flexible nonvolatile memories. Kim et al. [129] used graphene electrodes as the substrate for epitaxial growth of poly(vinylidene fluoride-trifluoroethylene) (PVDF-TrFE) polymer. The orientation of the crystals of PVDF-TrFE acquired a distinct symmetry due to the strong crystallographic interaction between the PVDF-TrFE and graphene. The orientation of the epitaxial PVDF-TrFE crystals was suitable for polarization switching in the presence of the electric field, leading to higher ferroelectric performance in metal/ferroelectric polymer/metal devices. m-poly(3,4-ethylene dioxythiophene (PEDOT) and poly(styrene sulfonic acid) (PSS) were used with graphene oxide for the fabrication of a transparent (70%) memory device by Shi et al. [130]. They fabricated an m-PEDOT: PSS/GO/m-PEDOT: PSS structure via a solution process where m-PEDOT: PSS films were doped with dimethyl sulfoxide. Spray coating was used to form m-PEDOT: PSS films, while GO film was prepared by spin coating, followed by a thermal treatment. The device revealed nonvolatile and volatile memory effects, with an ON/OFF current ratio of 10^4^ and 10^2^, respectively. The stable operation of the fabricated device and reliable switching endurance could pave the way for industrial-scale low-cost fabrication of future electronic devices. Boron nitride (BN) and graphene (G) were employed to fabricate an Au/Ti/G/BN/G/Au memristor by Zhu et al. [131]: this was able to switch between two or three resistive states. The switching between the numbers of states depended on the current limitation and reset voltage. The generation of the intermediate state resulted from the formation of a conductive nanofilament across the BN, as the graphene could limit the metal penetration. Polyimide-GO (PI-GO) nanocomposites with varied GO wt% (0, 0.1, 1.5, or 3 wt%) were synthesized by Choi et al. [132] to fabricate an Al/PI-GO(1.5%)/ITO resistive memory device with WORM switching behavior. The device depicted excellent characteristics, such as an excellent ON/OFF ratio (~10^8^), large retention time (~10^4^ s), good endurance (10^4^th cycle), and an outstanding yield (92%), with high transparency (82%). It was also observed that the performance of the memory device improved post-annealing.

### 3.4. Integrated Circuits with Graphene Transistors

Currently, for the fabrication of integrated circuits, complementary metal oxide semiconductor (CMOS) technology is used. However, the advancement of fabrication technology is approaching the limits of downsizing transistors in a rapid manner [133]. Graphene can be considered a potential substitute for post-Si electronics [134]. However, the fabrication technology of graphene-based low-power devices is still in the development stage, and an improved growth technique to produce large-area graphene films with superior electrical properties on dielectric surfaces needs to be developed (Figure 11). Meanwhile, Lemme et al. [135] reported the fabrication of a top-gated field-effect device using monolayer graphene. The carrier mobility value exceeded the universal mobility in the silicon and ultrathin body SOI MOSFETs. However, the authors suggested that bandgap tuning would be necessary to improve the device characteristics. Sun et al. [136] realized a double-gated GFET, with graphene as the working channel and organic ferroelectric sheets (PVDF-TrFE) as the top gating insulator. The inclusion of ferroelectric material substantially modified the charge transport properties in the graphene channels. The top-gate dielectrics isolated the graphene channels from the effects of the environment. Furthermore, the electrostatic-doping effects originating from the bottom gating were discovered to be an effective technique to modulate the doping level in the graphene channel and, thereby, to control the top-gated transfer characteristics of GFETs. An alternative design for an electrolyte-gated graphene FET (EGFET) configuration was proposed by Vieira et al. [137], where the source, drain, and gate would be in the same plane, so there was no need for an external gate electrode. Wafer-scale production of novel graphene EGFETs exhibiting carrier mobility up to 1800 cm^2^/Vs was possible because of the facile nature of this planar structure. They also prepared a chemical sensor as a proof of concept and demonstrated that the sensor could distinguish between different concentrations of saline liquids. Li et al. [138] developed graphene nanoribbons (GNRs) with a width below 10 nanometers, using them to fabricate field-effect transistor (FET)-like devices. They observed that the on–off current switching (I_on_/I_off_) at room-temperature could be induced by the gate voltage and increased exponentially as the GNR width decreased. Martini et al. [139] examined the relationships between the structure of the GNRs and the electrical properties of the GFET, made of those GNRs. Armchair-type GNRs of varying widths and lengths were used as the channel for a GFET, whereas graphene was used as the source and drain materials. They noticed that the output current values had a width-dependent characteristic, i.e., it depended on the magnitude of the electronic bandgap. A GNR-based transistor was also fabricated by Jangid et al. [140], with an I_ON_/I_OFF_ ratio of 2 × 10^7^, using nanoribbons and obtaining Pt-catalyzed etching of mechanically exfoliated graphene. The performance parameters were measured and the electron and hole mobilities were found to be 400 cm^2^/V.s and 1100 cm^2^/V·s, respectively, at 6 K. A graphene-based transistor for operating at high frequencies (gigahertz) was fabricated by Lin et al. [141] and frequency characteristics were analyzed by standard S-parameter measurements. A peak cut-off frequency of 26 GHz was measured for the graphene transistor, which had a gate length of 150 nm. The authors predicted that a cut-off frequency approaching a terahertz might be achieved for a GFET with a gate length of just 50 nm, provided that the high mobility of graphene could be preserved during the device fabrication process. Later, a graphene transistor with a higher operating frequency was synthesized by the same group [142] on a 2-inch graphene wafer.

Cut-off frequencies as high as 155 GHz were reported in a study by Wu et al. [143] on the properties of top-gated (gate length 40 nm) graphene r.f. transistors, with the frequency varying inversely to gate length. They also discovered that performance was not seriously impeded until the temperature reached 4.3 K, allowing for a wide operating window. Using solution-based graphene, Sire et al. [144] created a fast and flexible GFET with low resistance. The cut-off frequencies at an extrinsic and intrinsic current gain were 2.2 and 8.7 GHz, respectively, at a very low bias (VGS = 0.6 V and VDS = 0.65 V. The cut-off frequencies of the device power gains were measured at 550 MHz, indicating that the device might be employed in high-speed flexible electronics on plastic substrates. On flexible PI substrates, Lee et al. [145] constructed a GFET that achieved a high mobility of 3900 cm^2^/V.s at a 25 GHz cut-off frequency, and mechanical resilience down to a 0.7 mm bending radius. They further reported that, under exposition to DI water for short or long periods, the functional bilayer coating of the device, containing inorganic and organic thin films, formed a good hydrophobic surface that offered water-resistant protection, as well as reliable electrical performance. A flexible GFET having a gate length of 300nm was fabricated on PET substrates by Lan et al. [146]. A hole mobility of 1738 cm^2^/V∙s was achieved by coating the PET substrate with a Polyimide film, and Au-supported graphene-transfer technology facilitated the maintenance of the output resistance close to about 50 Ω. The modification in fabrication helped to realize an extrinsic f_max_ of 28 GHz. Graphene-based transistors often encounter the problem of high-emitter potential-barrier height during high-frequency applications. To overcome this limitation, Liu et al. [147] fabricated a vertical silicon–graphene–germanium transistor via the construction of a Schottky emitter between silicon and graphene. The Schottky emitter charging time was recorded as ≈118 ps, with a current value of 692 A/cm^2,^ and a capacitance of 41 nF/cm^2^, which indicated the feasibility of using the device in GHz-range applications. Montanaro et al. [148] fabricated high-speed optoelectronic mixers (OEM), employing a GFET for frequencies up to at least 67 GHz. On a single-layer graphene channel, the device could combine an electrical signal injected into the gate terminal with a modulated optical signal. The value of the Fermi level of graphene determined the photodetection process and the associated photocurrent sign. Liao et al. [149] devised an approach for the fabrication of high-speed graphene transistors with a self-aligned nanowire gate. The physical assembly of the nanowire gate maintained the high carrier mobility of graphene, while minimization of the access resistance was made possible by the self-alignment process, i.e., automatic and precise positioning of the source, drain, and gate electrodes. The fabricated transistor demonstrated a high intrinsic cut-off (transit) frequency, ranging from 100 to 300 GHz. In 2011, Lin et al. [150] reported the successful fabrication of a wafer-scale graphene circuit where all the components, specifically graphene field-effect transistors (GFETs) and inductors, were integrated on a single SiC wafer. The entire integrated circuit (IC), including the contact pads, was less than 1 mm^2^. The circuit was operated as a broadband Radio Frequency (RF) mixer with operating frequencies up to 10 GHz. Cheng et al. [151] devised a process for the scalable fabrication of self-aligned graphene transistors with transferred gate stacks on glass whose cut-off frequency was up to 427 GHz. However, the maximum oscillation frequency was only 29 MHz for a transistor with a 220 nm channel length. The authors indicated that maximum oscillation frequency could be further improved by increasing the graphene quality, reducing the gate resistance, and increasing the source–drain current saturation. Feng et al. [152] reported a fabrication process of ultra-clean self-aligned graphene transistors by pre-deposition of gold film on graphene as a protection layer. This method keeps graphene away from any possible contamination, resulting in good gate coupling and fewer parasitics, thus good dc and RF performances. The maximum oscillation frequency obtained from the 100 nm gate-length graphene transistor was 105 GHz. The fabrication of a graphene transistor on a flexible substrate (PET) for the active development of wearable electronic devices was demonstrated by Yeh et al. [153]. The device depicted an extrinsic cut-off frequency of 32 GHz and a maximum oscillation frequency of 20 GHz; when the device was subjected to a strain of 2.5%, the extrinsic cut-off frequency was 22 GHz and the maximum oscillation frequency was 13 GHz. Han et al. [154] reported a method to fabricate a high-performance three-stage graphene integrated circuit capable of working as a radio-frequency receiver, at a frequency of 4.3 GHz. The fabrication area of the integrated circuit was 0.6 mm^2^, which indicated the unprecedented graphene circuit complexity. The scaling of electronic devices is an important perspective for future electronics as it affects both the speed of the operation and packing density of IC technology. Bianchi et al. [155] studied the scaling of graphene ICs using transistors of varying gate lengths, from 3.3 to 0.5 μm, different channel widths, access lengths, and lead thicknesses. They found that the shortest gate delay of 31 ps per stage could be obtained in sub-micron graphene ring oscillators, oscillating at 4.3 GHz. Short-channel (20 nm) GFETs were fabricated by Llinas et al. [156] employing a thin, high K-gate dielectric and a 0.95 nm wide armchair-graphene nanoribbon as the channel material. At ambient temperature, the manufactured GFETs demonstrated excellent switching behavior, with a high I_ON_/I_OFF_ ratio of 10^5^. Hanna et al. [157] fabricated a 2.5 GHz GFET integrated power amplifier by thermal deposition on SiC. The power gain of the device was 8.9 dB, whereas the maximum reported output power and power-added efficiencies were 5.1 dBm and 2.2%, respectively. They also compared the performance of graphene and Si CMOS amplifiers, and concluded that the power-added efficiency was low compared with its the silicon counterpart. A graphene-based vertical hot-electron transistor was fabricated by Vaziri et al. [158] using a wafer-scale fabrication scheme compatible with silicon (CMOS) technology and its DC functionality was studied. Graphene was used as the base of the transistor and, upon application of an electric field, the state of the transistor was changed from ON to OFF, and vice versa, with an ON/OFF current ratio exceeding 10^4^. Liu et al. [159] fabricated a semiconductor–graphene–semiconductor transistor employing the Si membrane transfer method. The electrical characterization revealed that the common-base current gain could be improved by 10% compared with the standard gain. In a theoretical study, a graphene–molecule–graphene transistor design was proposed by Mol et al. [160]. The molecular junction insensitivity towards the atomic configuration of the graphene electrodes was attributed to the remarkably consistent single-electron charging observed. At ambient temperature, the graphene electrode stability allowed high-bias transport spectroscopy and the detection of numerous redox states. A large-area graphene ion-sensitive FET (ISFETs) was developed by Fakih et al. [161] for the simultaneous detection of different ions with high selectivity. The Nikolskii–Eisenman theory was employed by the authors to estimate the concentration of multiple ions with an LOD of 10^−5^ M from the values of the corresponding currents, while demonstrating the performance in an aquatic environment. Ning et al. [162] used CVD to grow a half-a-millimeter-long large-area single-crystal graphene (LSG) for use in the construction of a flexible ion gel-gated GFET on a PET substrate. They studied the electrical property of the device under bending and found that, at bending of 8.1%, the device could deliver a current on/off ratio of more than 400, as the graphene bandgap was expanded via deformation of the graphene crystal lattice. GFETs, made of chevron-type nanoporous graphene (NPG), were fabricated by Mutlu et al. [163] via bottom-up technology. With on–off ratios over 10^4^, the GFETs exhibited exceptional switching performance, which was interlinked with the structural quality of NPG. In air, the device worked as a p-type transistor, but when examined under vacuum, n-type transport was detected; this was related to the reversible adsorption of gases/moisture depicting the ballistic conductance nature and conduction anisotropy. Wu et al. [164] fabricated a vertical MoS_2_ transistor using the edge of a graphene layer as a gate electrode to achieve a gate length of less than 1 nm. The device featured subthreshold swing values as low as 117 mV/dec and On/Off ratios as high as 1.02 × 10^5^.

## 4. Critical Challenges of Graphene-Based Devices

High performance graphene-based electronic devices require good quality homogeneous graphene. However, the presence of different atomic-scale attributes, such as faults, impurities, disorders, grain boundaries, rotations, and anchoring groups hinders the goal of achieving large-scale graphene with a well-defined structure. Layer numbers, atomic orientation, ratio of C/O to C-C sp^2^, and lateral dimensions are also influencing factors which cause disparity in graphene grown by different methods.

The other major issue originates from the graphene transfer method, which is responsible for inducing damage, such as wrinkles, breakages, contaminants, etc., in the graphene structure. Thus, an efficient, non-invasive, large-scale transfer strategy is needed for efficient graphene production. Moreover, direct graphene growth on arbitrary substrates is another option for avoiding this problem; however, the graphene quality produced by this process is significantly inferior to that produced using metal catalysts. To generate transfer-free and high-quality graphene simultaneously, strategies, such as innovative growth processes, and advanced systems should be devised.

Controlling the structure and properties of graphene is linked to characterization approaches. Novel methodologies that are non-destructive, high precision, and fast are urgently needed to promptly analyze the atomic scale characteristics of large-area graphene, providing a foundation for graphene quality control. This is critical for effectively mapping or estimating the flaws, borders, and other aspects, to evaluate the application potential in specific sectors. The primary concerns, in terms of improving the effectiveness of most graphene-based devices, are the contact resistance of graphene, high sheet resistance, and work function, which play crucial roles in charge transfer. Modifications to substrates, graphene, and their interactions, will be the basic blueprints for overcoming these barriers.

The high cost of good quality graphene, which arises from the energy and substrate used, poses a barrier to its commercialization. The reuse of metal through optimal transfer techniques, and low temperature growth processes (e.g., cold-wall CVD), as well as effective collaboration with relevant sectors, might be advantageous in dealing with this challenge.

## 5. Summary and Future Outlook

The graphene “gold rush” began in 2004 when defect-free graphene was shown to have a wide range of exotic qualities, from mechanical to electrical. However, no developmental procedure for the growth of optimum-quality defect-free graphene with a large-area has yet been found, so the full potential of graphene is yet to be realized. Along with miniaturization, sustainability and eco-friendliness are important components in today’s society for the feasibility of any technological advancement, while graphene poses some serious health concerns due to its toxic nature. Thus, toxicity, reproducibility, a long shelf life, and sustainability are a few of the challenges that must be dealt with utmost care. These factors pose major hindrances to achieving graphene-based electronic applications (Figure 3).

Modern synthesis methods, such as CVD, show significant promise in producing large-area, (almost) defect-free graphene, paving the way for the fabrication of nanoelectronic devices. However, the marketability of a product is determined by the complexity of the synthesis procedure. Devices that demand the highest electronic-grade graphene, such as non-volatile memory, may require a longer time to emerge on a commercial basis, whereas graphene created via economical growth methods may be the first available in the market.

More research should focus on the use of graphene in batteries, including functional modification of graphene for increased wetting of thick electrodes, separator modification through proper use of rGO, prevention of graphene restacking during electrode fabrication, and inhibition of graphene absorption of the electrolyte during battery operation. The development of large-area graphene is essential for producing transparent electronics based on graphene at a reasonable cost, and defect-free criteria are the most significant aspect in the case of the GFET type of structure.

Given recent advancements that show a sharp decline in the cost of producing high-quality graphene, the material has enormous potential from an economic standpoint. In a nutshell, rapid progress toward defect-free, large-area, sustainable graphene production for different applications is required, but health concerns must also be addressed.

The authors believe that graphene-based electronic devices must be at peerless level in both cost and performance in order to replace the present Si-based technology. Moreover, the cost, graphene characteristics, and feasibility of graphene-based electronic devices are all interconnected. As a result, for each application, a balance of these three criteria is essential. The decreasing cost of high-quality graphene sheets will accelerate the commercialization of high-performance materials as manufacturing procedures and equipment become sufficiently mature.

## Data Availability

Not applicable.

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
