# Peer review of "Graphene-Induced Performance Enhancement of Batteries, Touch Screens, Transparent Memory, and Integrated Circuits: A Critical Review on a Decade of Developments"

_nanomaterials, 2022, doi:10.3390/nano12183146_

Round 1
Reviewer 1 Report
I propose to the authors to realize the next changes:
1. Improve the text on page 4.
2. In Fig. 6 change the order of the valence and conduction bands.
3. Add the list of all abbreviations used.
Author Response
Reviewer 1
Comment 1. Improve the text on page 4.
Response: The text has been improved as suggested.
Comment 2. In Fig. 6 change the order of the valence and conduction bands.
Response: The order of the valence and conduction bands has been changed. In the old figure, valance band was at the top and the conduction band was shown at the bottom. Now the position has been reversed as suggested.
Comment 3. Add the list of all abbreviations used.
Response: An abbreviation list has been added as suggested.

Reviewer 2 Report
In this work, Hussain et al. reviewed the structure and properties of graphene materials and summarized their applications in electronic devices such as batteries, touch screens, transparent memory, and integrated circuits. The manuscript the comprehensive and well organized. I think it can be accepted to publish on Nanomaterials after addressing the following issues.
1. In each application (especially batteries), a number of literature results are listed. The reviewer believes that it is better to build tables to put these data, which is easier for the readers to compare.
2. Inconsistent reference numbers in the main text and reference sections.
3. There are many typos that need to be fixed.
Author Response
Reviewer 2
Comment 1. In each application (especially batteries), a number of literature results are listed. The reviewer believes that it is better to build tables to put these data, which is easier for the readers to compare.
Response: A tabular representation depicting the different electrode materials used for capacity enhancement of LIB has been added as suggested.
Comment 2. Inconsistent reference numbers in the main text and reference sections.
Response: Reference numbers are made consistent as suggested.
Comment 3. There are many typos that need to be fixed.
Response: The typos have been fixed as suggested.

Reviewer 3 Report
The manuscript describes recent progress about graphene-based devices in the application of batteries, touch screens, transparent memory and integrated circuits. The manuscript is timely and has been well organized. Therefore, I consider that this manuscript has a value to be published in Nanomaterials with a minor revision.
1. It would be better to deeply discuss the critical challenges of graphene-based devices before the section of “Summary and Future Outlook”.
2. The serial numbers of references are digital gibberish in the PDF manuscript. Please double check it.
Author Response
Reviewer 3
Comment 1. It would be better to deeply discuss the critical challenges of graphene-based devices before the section of “Summary and Future Outlook”.
Response: A section has been added on the critical challenges of graphene-based devices before the section of “Summary and Future Outlook”. Consequently, we have also modified the “Summary and Future Outlook” section as recommended.
Comment 2. The serial numbers of references are digital gibberish in the PDF manuscript. Please double check it.
Response: Reference numbers are checked and made consistent as suggested.

Reviewer 4 Report
In this manuscript, the authors present a critical review on the progress of graphene research on the performance enhancement of electronic devices, such as batteries, displays and integrated circuits. Considering the importance of such rapidly growing field, this review paper would attract much attention from the communities of both electronic devices and two-dimensional materials. Therefore, I would suggested its acceptance after addressing the following concerns:
1. The authors shall introduce some typical methods for the production of graphene with diverse dimensions and quality, and the selection rules of graphene materials for the design and applications of energy storage and other electronic devices.
2. I would expect some personal opinions from the authors on the relations between material characteristics, devices structures and the performance.
3. It would be helpful if the authors could provide some conceptual sketches in the section of outlook.
4. Please check the typos in the manuscript, such as the spanish names in Ref. 8 and Ref. 34. Moreover, the authors shall combine some figures in the same category (e.g. Figure 1 to 4) into a bigger one.
Author Response
Reviewer 4
Comment 1. The authors shall introduce some typical methods for the production of graphene with diverse dimensions and quality, and the selection rules of graphene materials for the design and applications of energy storage and other electronic devices.
Response: A new scheme (Scheme 2) has been incorporated into the modified manuscript to depict the different synthesis methods, dimensions, and quality of grown samples along with their target application fields. Moreover, the selection rules for mapping graphene properties with potential applications in electronic devices have already been shown in Scheme 1 as suggested.
Comment 2. I would expect some personal opinions from the authors on the relations between material characteristics, devices structures and the performance.
Response: Authors believe that, graphene-based electronic devices need to be at peerless level in both cost and performance in order to replace the present Si-based technology. Moreover, the cost, graphene characteristics, and feasibility of graphene-based electronic devices are all interconnected. As a result, for each application, a balance of these three criteria is essential. The decreasing cost of high-quality graphene sheets will speed up the commercialization of high-performance materials as manufacturing procedures and equipment get mature enough as suggested.
Then we have included the above paragraph in the “Summary and Future Outlook” section.
Comment 3. It would be helpful if the authors could provide some conceptual sketches in the section of outlook.
Response: A conceptual sketch has been added in the “Summary and Future Outlook” section as suggested.
Comment 4. Please check the typos in the manuscript, such as the spanish names in Ref. 8 and Ref. 34. Moreover, the authors shall combine some figures in the same category (e.g. Figure 1 to 4) into a bigger one
Response: Refs 8 and 34 have been re-verified and found correct. However, there were many typos in the old manuscript, and those have been corrected as suggested.
Then, we combined Figures 1 to 4 into a single bigger figure following the suggestion of the reviewer.
